# Tumor-Specific Immunoenhancing Effects after Local Cryoablation for Metastatic Bone Tumor in a Mouse Model

**DOI:** 10.3390/ijms23169445

**Published:** 2022-08-21

**Authors:** Ryohei Annen, Satoshi Kato, Satoru Demura, Shinji Miwa, Akira Yokka, Kazuya Shinmura, Noriaki Yokogawa, Noritaka Yonezawa, Motoya Kobayashi, Yuki Kurokawa, Toshifumi Gabata, Hiroyuki Tsuchiya

**Affiliations:** 1Department of Orthopaedic Surgery, Graduate School of Medical Sciences, Kanazawa University, 13-1 Takara-machi, Kanazawa 920-8641, Japan; 2Department of Radiology, Graduate School of Medical Sciences, Kanazawa University, 13-1 Takara-machi, Kanazawa 920-8641, Japan

**Keywords:** abscopal effect, cryoablation, metastatic bone tumor, immunoenhancing

## Abstract

We investigated the abscopal effect after cryoablation (CA) on bone metastasis using a mouse model. Breast cancer cells were implanted in the bilateral tibiae of mice. The left tumor was treated locally with CA, and the right abscopal tumor (AT) was left untreated. The mice were divided into four groups based on the combination of CA and intraperitoneal administration of anti-PD-1 antibody (PD) as treatment interventions (Control, CA, PD, and CA + PD). The reduction ratio of the size of AT, the quantitative immune effects at enzyme-linked immunospot (ELISPOT) assay, and the intensity of infiltration of immune-related cells to AT were compared among the groups. CA alone showed a significant immunoenhancing effect on the volume change ratio of AT from day 0 to day 14 (Control-CA: *p* < 0.05), ELISPOT assay (Control-CA: *p* < 0.01), and CD4^+^ cell count in immunostaining (Control-CA: *p* < 0.05). CA alone showed no significant immunoenhancing effect on CD8^+^ and Foxp3^+^ cell counts in immunostaining, but the combination of CA and PD showed a significant immunoenhancing effect (Control-CA + PD: *p* < 0.01 [CD8, Foxp3]). The results suggested that the abscopal effect associated with the local cryotherapy of metastatic bone tumors was activated by CA and enhanced by its combination with PD.

## 1. Introduction

The skeleton is the third most common site of metastasis after the lungs and liver, with the spine being the most frequently affected bone [1]. Spinal metastases occur in 20–40% of patients with cancer, and up to 20% of them become symptomatic [2,3,4]. Spinal metastases often cause severe pain and neurological symptoms, which reduce the performance status (PS) and increase patient mortality [5,6]. In patients with metastatic cancer, a lowered PS caused by spinal lesions directly and indirectly affects mortality by hampering the application of systemic therapies. Although these patient conditions require local therapies to improve or maintain PS, most patients with spinal metastases also have multiple metastases in other organs or other skeletal sites. Therefore, when applying local therapies for bone metastases, especially in the spine, it would be ideal to append systemic anti-cancer effects or potentiate systemic cancer therapies.

Shrinkage of untreated distant metastases after local therapy has been observed, which is referred to as the abscopal effect. The abscopal effect was originally mentioned by Mole et al. in 1953 for the phenomenon of shrinkage of untreated distant metastases after radiation therapy (RT) [7]. Recently, this phenomenon was proposed as a cancer-specific immunoenhancing effect triggered by tumor cell death [8], and the abscopal effect has been reported in various local therapies, such as RT, embolization, radiofrequency ablation (RFA), and cryotherapy [9,10,11,12]. In particular, cryotherapy is an area of current interest to us, as previous studies have investigated the immune elevating effects associated with frozen autografts [13].

Cryoablation (CA) is a type of cryotherapy performed for solid tumors, such as renal cancer. CA is less painful than RFA, minimally invasive [14], and effective even for tumor refractory to radiotherapy [15]. As a result of these advantages, CA is now being considered for clinical applications in bone metastases [16,17].

On the other hand, the abscopal effect associated with CA on bone metastases has not yet been fully verified, and its applicability to the skeletal system, which often involves multiple lesions, is unknown. Furthermore, in the clinical field, there is a prevailing opinion that it is difficult to obtain a sufficient abscopal effect by local therapy alone, and methods to enhance the therapeutic effect using adjuvants, such as anti-PD-1 antibodies (PD), have been advocated [12,18]. Against this background, we aimed to verify the abscopal effect associated with CA using a mouse model of bone metastasis, which has never been reported before, and to investigate the synergistic effect of CA with immune checkpoint inhibitors.

## 2. Results

### 2.1. Abscopal Tumor Growth

The mean volume of the abscopal tumor (AT) (mean ± SD) in the Control, CA, PD, and CA + PD groups (*n* = 8 in each group) was 0.87 ± 0.24, 0.75 ± 0.38, 0.99 ± 0.23, and 1.09 ± 0.19 cm^3^, respectively, at day 0 (10 days after tumor inoculation); 1.20 ± 0.37, 0.46 ± 0.11, 0.49 ± 0.18, and 0.44 ± 0.07 cm^3^, respectively, at day 7; and 1.57 ± 0.52, 0.35 ± 0.14, 0.36 ± 0.13, and 0.32 ± 0.05 cm^3^, respectively, at day 14. The volume change ratio of AT in the Control, CA, PD, and CA + PD groups from day 0 to day 7 was 1.43 ± 0.37, 0.74 ± 0.30, 0.50 ± 0.15, and 0.41 ± 0.09, respectively, and from day 0 to day 14 was 1.84 ± 0.40, 0.47 ± 0.17, 0.37 ± 0.12, and 0.30 ± 0.05, respectively (Figure 1). From day 0 to day 7, the PD and CA + PD groups showed significant tumor shrinkage compared to that in the Control group (*p* < 0.01), whereas from day 0 to day 14, the CA, PD, and CA + PD groups all showed significant tumor shrinkage compared to that in the Control group (Control-CA: *p* < 0.05, Control-PD, Control-CA + PD: *p* < 0.01) (Figure 1a,b).

### 2.2. Spleen Weight

As a correlation between splenomegaly and the development of tumor size has been reported [19], we also investigated the spleen weights after sacrificing the mice to obtain reference values. Representative examples of spleen weight and size are shown in Figure 2a–c. A positive correlation was observed between the spleen weight and AT size on day 14. The relationship between variables was expressed with the equation y = 1.73x − 0.02, showing a significant correlation with a Spearman correlation coefficient of 0.70 (Figure 2d). From this correlation, the validity of the AT size, which was measured by calipers and calculated by an approximate formula, was objectively supported by the spleen weight. The mean weight of the spleen (mean ± SD) in the Control, CA, PD, and CA + PD groups (*n* = 8 in each group) was 0.91 ± 0.21, 0.17 ± 0.052, 0.29 ± 0.13, and 0.20 ± 0.080 g, respectively (Figure 2e). Tumor size was significantly smaller in the other groups (CA, PD, CA + PD) than in the Control group (*p* < 0.01) (Figure 2f).

### 2.3. Enzyme-Linked Immunospot (ELISPOT) Assay

After sacrificing the mice on day 14, we performed an enzyme-linked immunospot (ELISPOT) assay, which quantifies the number of T cells that specifically react to tumor antigens and secrete INF-γ as spots. A lot of cytokines are involved in the regulation of PD-1. However, as INF-γ is one of the most representative immune cytokines associated with the regulation of PD-1 [20], we investigated immune responses specifically to INF-γ. A higher number of spots indicated a higher tumor-specific immune response. Figure 3a depicts the representative findings of the ELISPOT assay. The mean number of spots (mean ± SD) in the Control, CA, PD, and CA + PD groups (*n* = 8 in each group) was 31.6 ± 9.5, 107.3 ± 45.0, 46.9 ± 14.0, and 123.3 ± 42.2, respectively. Compared to the Control group, the CA and CA + PD groups showed a significant increase in the number of spots (*p* < 0.01). In addition, there was a significant difference in the number of spots between the PD and CA + PD groups (*p* < 0.05) (Figure 3b).

### 2.4. Immunohistochemistry

After sacrificing the mice on day 14, The right lower limb containing AT was harvested, and hematoxylin-eosin (H-E) staining and immunostaining for CD4, CD8, and Foxp3 were performed (Figure 4). The mean number of immune-related T cells infiltrating the ATs (mean ± SD) was observed under a high-power field (HPF). The Control, CA, PD, CA + PD groups (*n* = 8 in each group) had 1.25 ± 0.89, 18.38 ± 5.53, 22.17 ± 5.10, and 29.13 ± 10.42 CD4^+^ T cells; 2.23 ± 1.06, 9.62 ± 2.17, 11.0 ± 3.02, 26.4 ± 9.70 CD8^+^ T cells; and 12.7 ± 6.83, 5.65 ± 2.22, 2.65 ± 1.86, 2.08 ± 1.49 Foxp3^+^ T cells, respectively. The number of CD4^+^ cells was significantly higher in the CA (*p* < 0.05), PD (*p* < 0.01), and CA + PD (*p* < 0.01) groups than that in the Control group. The number of CD8^+^ cells was significantly higher in the PD (*p* < 0.05) and CA + PD (*p* < 0.01) groups than that in the Control group, and the number of positive cells was also significantly higher in the CA + PD (*p* < 0.01) group than that in the CA group. The number of Foxp3^+^ cells was significantly decreased in the PD (*p* < 0.01) and CA + PD (*p* < 0.01) groups compared to that in the Control group (Figure 5). Representative images of the CD4^+^, CD8^+^, and Foxp3^+^ cells in each group are shown (Figure 6).

## 3. Discussion

Over the past several decades, cryotherapy has been used to treat malignant tumors of the skin, prostate, liver, breast, lung, and bone, and many more applications have been investigated. Compared to surgical resection, cryotherapy has several advantages, including minimally invasive treatment, less damage to the surrounding structures, and less pain due to the anesthetic effect of freezing [21]. Recently, CA, a type of cryotherapy, has attracted considerable attention. CA is a local therapy in which a probe is percutaneously inserted into the body to necrose tumor tissue with real-time monitoring of “ice-ball: the area of freezing” via imaging; this treatment has been clinically applied to kidney cancer and other organ tumors [22], and its clinical application to bone metastases, including the spine, is being considered [16,17]. As cryotherapy, including CA, is used clinically, another potential advantage attracting attention is the shrinkage effect of untreated distant metastases identified after the cryotherapy of the primary tumor, that is, the abscopal effect. This suggests the possibility of a systemic therapeutic effect of local cryotherapy [23,24,25]. As immune checkpoint inhibitors, such as anti-PD-1 antibodies, have been adopted into clinical use, the abscopal effect associated with CA has attracted more attention in terms of its synergistic effects with adjuvants. For the clinical treatment of bone metastases, it is important to determine the degree of abscopal effect that can be expected from CA alone and the practical anti-tumor effect expected from the adjuvant combination. Therefore, we attempted to verify the abscopal effect associated with CA and the synergistic effects of immune checkpoint inhibitors in a mouse model of bone metastasis, which has never been reported. The important points of the present results are that the immunoenhancing effect obtained with CA alone was partially larger than expected and that the significant immunoenhancing effect was confirmed with the PD combination even in the results where no significant immunoenhancing effect was confirmed with CA alone.

The three endpoints in this study were AT size change ratio, ELISPOT assay, and immunostaining. A reduction in AT size ratio suggests a reduction in distant tumor size. The ELISPOT assay detects the number of mice splenocytes that specifically reacted to tumor antigen and secreted cytokines, suggesting an enhanced immune response. In immunostaining, we stained three cell types. The immune system has multiple subsets, and interactions between these subsets are essential for maximizing immune responses against infection and tumors [26]. To analyze the trends of these subsets from multiple angles, the number of T cells with subsets of CD4^+^, CD8^+^, and Foxp3^+^ cells infiltrating the AT was investigated. CD4^+^ T cells are key orchestrators of adaptive immune responses [27] and are one of the starting points for various cascade reactions and the regulation of immune responses. CD8^+^ T cells primarily destroy malignant cells, such as cytolytic T lymphocytes (CTLs) [28]. Foxp3 is a marker of regulatory T cells (Tregs). Tregs suppress the proliferation and function of effector T cells, and cancer cells may thereby escape their cytotoxic effects [29,30]. Based on the above, CD4^+^ and CD8^+^ T cells were used as indicators of the upregulation of anti-tumor effects, while Foxp3^+^ T cells were used as indicators of downregulation.

Significant immune upregulation by CA alone was obtained in three areas: tumor size change ratio on day 14, ELISPOT assay, and CD4^+^ cell counts in immunostaining. There have been scattered reports that other local therapies, such as RFA, do not provide sufficient immunoenhancing effects [31,32]. In contrast, compared to other local therapies (RFA, irreversible electroporation, etc.), CA is more likely to induce tumor-specific T cell activation by antigen presentation because the tumor antigen is less denatured [33] and the antigen is slowly exposed to the bloodstream during the thawing process after freezing, resulting in a better immune response [34]. As in this experiment, previous studies of CA in a mouse model of hepatocellular carcinoma reported significantly increased CD8^+^ T cell infiltration into distant tumors on days 3 and 7, suggesting an abscopal effect of CA alone [35], implying that CA has a strong abscopal effect. We believe that the present results are mostly consistent with these reports and also partially suggest an immunoenhancing effect of CA in the metastatic bone tumor model. In contrast, CD8^+^ and Foxp3^+^ cell counts in immunostaining were not significantly increased by CA alone. CD8^+^ and Foxp3^+^ were not significantly different from the Control group in CA alone but were significantly different from the Control group in the CA + PD group. Furthermore, there was a significant difference in the number of CD8^+^ cells between the CA and CA + PD groups. These results suggest that the combination of CA and PD enhanced the immune effect in a subset which were not significantly immunocompetent with CA alone. In a basic experiment using a mouse model of renal carcinoma, CA combined with anti-PD-1 antibody demonstrated a greater immunoenhancing effect than that of the anti-PD-1 antibody alone [35], and in a clinical trial of basal cell carcinoma, the combination of imiquimod (TLR7/8 agonist) and CA was associated with a five-year recurrence-free survival rate of 91% [36]. Both basic and clinical studies have shown that the combination of CA and adjuvant has a synergistic immunoenhancing effect, which we believe was demonstrated in this experiment. Various definitions have been proposed for the synergistic effect [35]. In this study, if we adopt the definition of the dose equivalence principle (DEP) [37], proposed by Loewe et al. in 1926, which states that the results of each group exceed the additive effect, we consider that synergistic effects due to CA and PD were obtained in the CD8^+^ and Foxp3^+^ results. In contrast, the synergistic effect of CA and PD was not confirmed in the AT size change ratio and ELISPOT assay. In the AT size change ratio, it is presumed that the anti-tumor effect of the anti-PD-1 antibody was too strong, and the tumor size reduction almost reached a plateau before the endpoint, so the comparison of AT size change ratio did not confirm the synergistic effect. In the ELISOPT assay, the PD group showed poor immunoenhancement, and the increase in immunity in the CA + PD group was only additive, not synergistic. The reason for this is unclear; however, studies examining the synergistic effect of RFA’s abscopal effect and anti-PD-1 antibody in a mouse prostate cancer model [26] and carboplatin and anti-PD-1 antibody in a mouse breast cancer model [38] showed that anti-PD-1 antibody alone did not significantly increase the number of spots, which is consistent with the results of the present study in that no significant increase in the number of spots was observed with PD alone.

Since the results of the present study suggest an abscopal effect of CA alone and a synergistic effect of CA and PD combination for clinical application in the future, it is desirable to demonstrate the immunological efficacy of CA in comparison with other local therapies. In addition, the focus should be on specific conditions conducive to the abscopal effect, that is, the adjustment of cryo-techniques (freezing frequency and dose and timing of anti-PD-1 antibody administration) and the validation of approaches to avoid side effects and maximize efficacy. We hope that this research will lead to future clinical applications in humans. Spinal metastases often require surgical treatment due to severe pain, neurological symptoms, or both. Although CA is applicable alone as a local therapy, it can be safely used in combination with surgical spinal cord decompression and stabilization. Most patients who undergo surgery for symptomatic spinal metastases have other organ lesions. We believe that hybrid surgery combined with CA can not only improve symptoms and local tumor control but also provide the activation of systemic tumor immunity. Furthermore, adjuvants such as PD will enhance the systemic therapeutic effect. We believe that this study, which verified the abscopal effect on bone metastasis of cryoablation, could be the first step toward further clinical application in humans.

The current study had several limitations. Our mouse model of bone metastasis was not a bone metastasis model by strict definition. Tumor metastasis involves a series of sequential steps, such as local invasion to escape from the surrounding tissues of the primary tumor, the invasion of the blood or lymphatic vessels, survival in the circulation, escape from the circulatory system, extravasation, adaptation to the microenvironment, and transformation to form a metastatic lesion [39]. However, it is practically impossible to produce this phenomenon at specific sites in the mouse body, particularly in the skeletal system. Therefore, we used an alternative method of injecting breast cancer cells directly into the tibia. Nude mice were not used in this experiment. Therefore, the effect of autoimmune rejection could not be eliminated. However, we considered that this experimental method is appropriate because there are precedents for similar experimental systems in previous reports [32,40]. Cell counting by fluorescence-activated cell sorting (FACS) is preferable to that by immunostaining, but FACS was not available in our institution. Therefore, we adopted evaluation with cell counting in immunostaining.

## 4. Materials and Methods

### 4.1. Animals

Female C3H mice were obtained from Sankyo Labo, Inc. (Toyama, Japan). This study was approved by the Committee of Animal Care and Experimentation of Kanazawa University (Kanazawa, Japan, AP-194053). The mice were maintained in cages (five mice per cage) under natural illumination, with free access to water and nutrients. The mice were anesthetized via intraperitoneal injection using a mixture of medetomidine 0.75 mg/kg, midazolam (4.0 mg/kg), butorphanol (5.0 mg/kg), and saline solution. All surgeries were performed under anesthesia, and efforts were made to minimize the suffering of the mice. Following each procedure, the mice were monitored in their cages until they recovered from anesthesia. The health of the mice was monitored daily during the study. The mice were euthanized by an intraperitoneal overdose injection of pentobarbital sodium.

### 4.2. Tumors

The murine breast cancer cell line, MMT-060562, derived from a spontaneous mammary tumor in a C57BL and A/F1 hybrid female mouse, was provided by the European Collection of Authenticated Cell Cultures Cell Lines, Australia. These cells were maintained in a complete medium consisting of Dulbecco’s Modified Eagle Medium (high glucose) with L-glutamine and phenol red (Wako Pure Chemical Industries, Osaka, Japan) supplemented with 10% heat-inactivated fetal bovine serum, 100 μg/mL of streptomycin, and 100 units/mL of penicillin, and cultured at 37 °C in 5% CO_2_. The breast cancer cell line MMT-060562 was used because it is a naturally occurring breast cancer cell line that can easily generate bone metastasis models in non-immunodeficient mice, making it useful for immunological validation [41,42].

### 4.3. Freezing Device

On the basis of a previous report [43], a freezing device was created for our improvement. A 10 mm syringe was incorporated into a 20-cc disposable syringe to create a layer of air and prevent heat conduction from the syringe surface, and a 1.2 mm diameter copper wire was placed at the tip of the syringe to form a cryoprobe (Figure 7a,b). By injecting liquid nitrogen into the device, the tip of the copper wire was rapidly cooled via thermal conduction. In a preliminary experiment, it was confirmed that an area with a radius of 5 mm from the tip of the probe was quickly cooled to −50 °C or lower, the temperature required to kill malignant cells in vivo [44]. Based on previous reports [15,40], two cycles of 60 s freezing and 60 s thawing at the room temperature of 25 °C were performed, which are considered to easily produce immunological effects.

### 4.4. Procedure

To establish a mouse model of bone metastasis, a small 5 mm skin incision was made in the bilateral lower limbs of the mouse, a puncture hole was made on the tibial tuberosity with an 18G needle, and the intrathecal cavity was drilled with a 23G needle. Mouse mammary tumor (MMT) cells 3 × 10^6^ MMT were suspended in 100 μL of Matrigel (BD Biosciences, San Jose, CA, USA), and 5 μL of the solution was injected into each tibia (Figure 8a). The right tumor was left untreated to serve as the control for evaluation and was designated as the AT, whereas intervention (CA and/or intraperitoneal administration of anti-PD-1 antibody (PD)) was applied to the left tumor, with CA performed ten days after tumor implantation. Four groups were established according to the combination of interventions (*n* = 8 in each group, 32 mice in total). The Control group only underwent skin incision and probe insertion as a sham operation, the CA group underwent CA, the PD group underwent sham operation and PD, and the CA + PD group underwent both CA and PD. The AT size was measured on days 7 and 14 by measuring the length, width, and height using calipers, and tumor size was determined using an approximation formula (length × width × height × π/6) based on previous reports [45]. On day 14, the mice were euthanized as prescribed, and the right lower limbs containing the AT and spleen were sampled. After weighing the spleen, splenocytes were used for the quantitative evaluation of tumor-specific immunoenhancing effects by ELISPOT assay, and the AT was immunostained to evaluate the degree of immune-related cell infiltration.

### 4.5. Immune Checkpoint Inhibitor for Adjuvant Treatment

Anti-mouse PD-1 (CD279) (Leinco Technologies, Inc., Fenton, MO, USA) was intraperitoneally injected into the mice of the applicable groups on days 0 (20 mg/kg), 5 (10 mg/kg), and 10 (10 mg/kg) (Figure 8b).

### 4.6. Evaluation of the Spleen Weight

The mice were euthanized 14 days after the first intervention. The spleen was excised and its weight was measured using an electronic scale (Mettler Toledo, Tokyo, Japan).

### 4.7. Tumor-Specific IFN-γ ELISPOT Assay

The number of tumor-specific IFN-γ-producing splenocytes was measured using the ELISPOT assay. A 96-well plate was coated with an anti-mouse IFN-γ antibody (Cellular Technology Ltd., Cleveland, OH, USA). Activated splenocytes (1 × 10^5^ cells/well) were cultured for 24 h at 37 °C in a 5% CO_2_ incubator alone or in the presence of MMT-060562 tumor cells (5 × 10^4^ cells/well). Thereafter, the wells were washed and incubated with a biotinylated anti-IFN-γ antibody. After the reactions were visualized, the spots were counted using anti-biotin-AP after 24 h of incubation at 37 °C via an immune-spot analyzer (Cellular Technology Ltd., Cleveland, OH, USA), and the results were expressed as the number of cytokine-producing cells. The average number of spots in three wells was treated as the number of reaction spots for each sample. The ELISPOT assay was performed for the Control, CA, PD, and CA + PD groups.

### 4.8. Evaluation of the Infiltration Intensity of Immune-Related Cells in the ATs

Immunohistochemistry for CD4, CD8, and Foxp3 was performed as follows. Each tissue sample from the right lower limb containing the AT was embedded in paraffin and sectioned at 4 μm thickness. The following primary antibodies were used: anti-CD4 rabbit polyclonal antibody (Bioss, Woburn, MA, USA; bs-0647R; 1:200), anti-CD8 rabbit polyclonal antibody (Biorbyt Ltd., Cambridge, UK; orb10325; 1:100), and anti-Foxp3 rabbit polyclonal antibody (Novus Biologicals, Colorado, USA; NB100-39002; 1:200). Anti-mouse or rabbit IgG conjugated with peroxidase-labeled polymers (EnVision, Shanghai, China; Dako, Carpinteria, CA, USA) was used as the secondary antibody. The number of infiltrating immune-related cells surrounding the ATs was measured in the Control, CA, PD, and CA + PD groups (*n* = 8 for each group). Photographs of the ATs were captured from the four groups. For each analysis, five independent HPFs of the AT region were randomly selected per mouse. Observations were performed using a BZ-9000 microscope (KEYENCE, Osaka, Japan). The slides were coded and the positive cells for each marker were counted by two examiners who were blinded to the slides. All the mice were evaluated with the same resolution and clarity. The mean value of the positive cells in one HPF of each mouse was summed and divided by the total number of mice to obtain the mean value and standard deviation.

### 4.9. Statistical Analysis

Mean differences between the groups were statistically evaluated using the Kruskal–Wallis test, followed by the Bonferroni test. Statistical significance was set at *p* < 0.05 (* *p* < 0.05, ** *p* < 0.01). Statistical analysis was performed using the SPSS statistical software version 27 (SPSS, Chicago, IL, USA).

## 5. Conclusions

The abscopal effect of cryoablation was confirmed in a mouse model of metastatic bone tumors. The combination of cryoablation and immune checkpoint inhibitors enhances the abscopal effect of cryoablation.

## Figures and Tables

**Figure 1 ijms-23-09445-f001:**
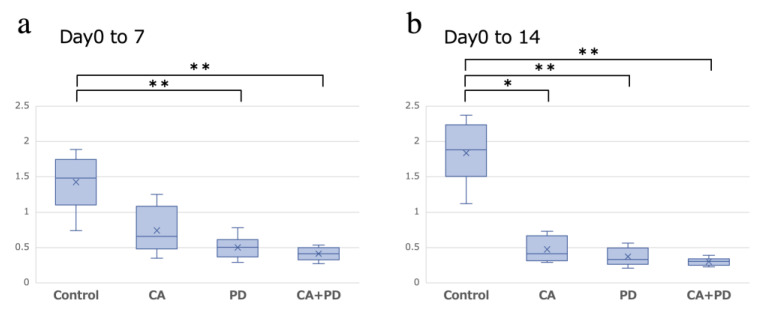
The volume change ratio of AT from day 0 to day 7 and from day 0 to day 14. (**a**) From day 0 to day 7, there were no significant differences between the Control and CA groups; however, there were significant differences between the Control and PD groups and between the Control and CA + PD groups (** *p* < 0.01). (**b**) From day 0 to day 14, there was a significant difference between the Control group and the other three groups (* *p* < 0.05, ** *p* < 0.01); however, we found no significant difference between the CA and anti-PD-1 antibody treatment groups.

**Figure 2 ijms-23-09445-f002:**
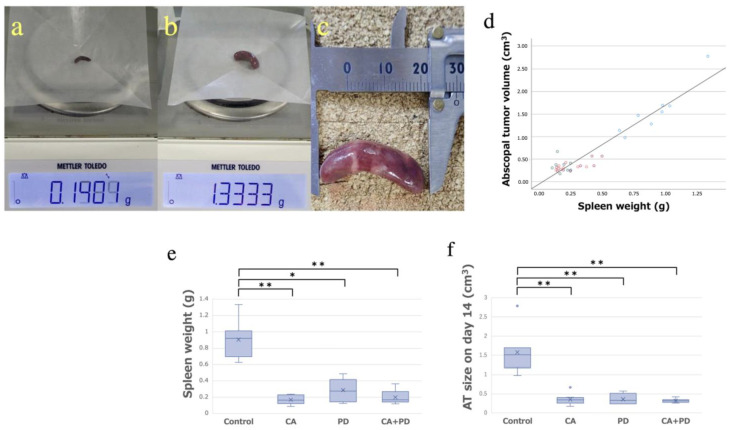
(**a**) Spleen weight of the No. 1 sample in the CA + PD group. (**b**,**c**) Spleen weight and size of the No. 4 sample of the Control group in splenomegaly condition. (**d**) Correlation between the spleen weight and AT size. A significant correlation was observed, with a Spearman correlation coefficient of 0.70. (**e**,**f**) There was a similar trend between the spleen weight and the AT size as there was a significant difference between the Control and other groups (* *p* < 0.05, ** *p* < 0.01).

**Figure 3 ijms-23-09445-f003:**
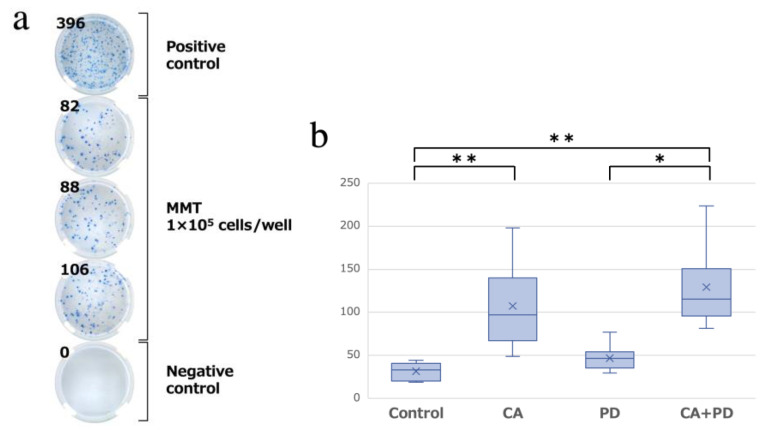
(**a**) Representative findings of the well design. The result of the No. 3 sample in the CA group. (**b**) The results of the ELISPOT assay. There was a significant difference in the number of spots between the Control group and the CA, CA + PD groups (** *p* < 0.01). There was also a significant difference in the number of spots between the PD group and the CA + PD group (* *p* < 0.05).

**Figure 4 ijms-23-09445-f004:**
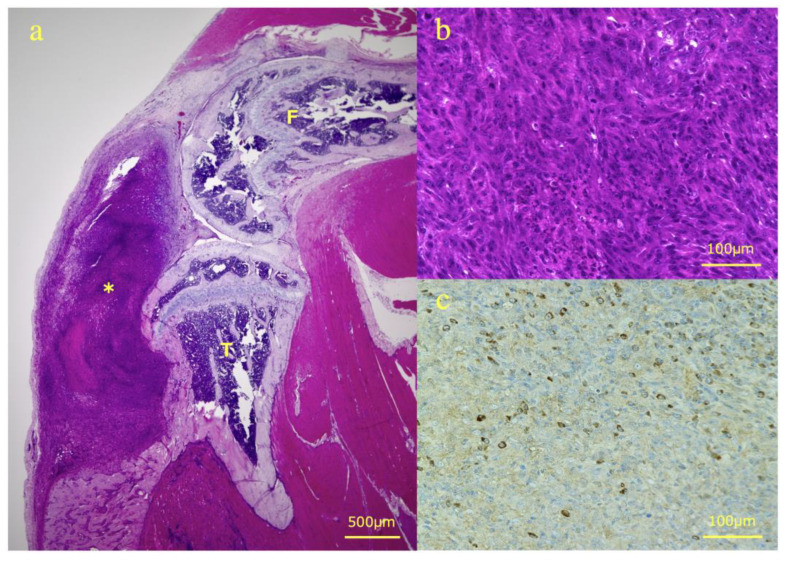
Representative findings of H-E staining and immunostaining. (**a**) H-E staining of a sagittal slice of the right lower limb. AT in the tibia is extending extraosseous (F: femur, T: tibia, *: AT). (**b**) HPF of the AT tissue with H-E staining. (**c**) HPF of AT tissue of the No. 3 sample in the CA group with CD8 immunostaining. Brown-stained cells are positive. HPF, high-power field; AT, abscopal tumor.

**Figure 5 ijms-23-09445-f005:**
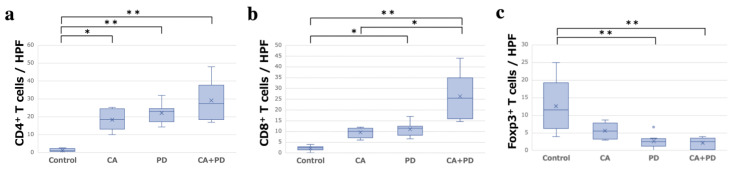
Number of immune-related cells infiltrating the abscopal tumor observed under HPF in the four groups (* *p* < 0.05, ** *p* < 0.01). (**a**) Number of CD4^+^ T cells in the four groups. (**b**) Number of CD8^+^ T cells in the four groups. (**c**) Number of Foxp3^+^ T cells in the four groups.

**Figure 6 ijms-23-09445-f006:**
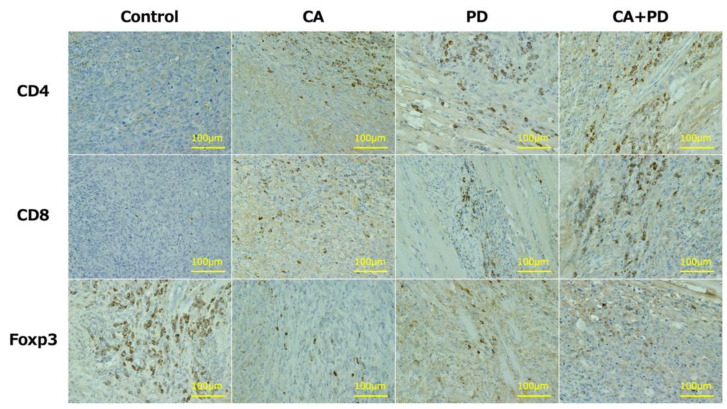
Representative findings of immunohistochemistry for CD4, CD8, and Foxp3 in each group. Expressions reveal a cytomembrane pattern.

**Figure 7 ijms-23-09445-f007:**
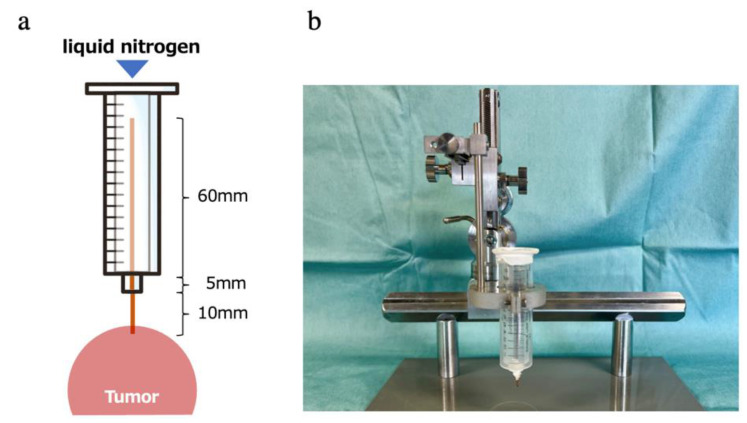
A device for CA that we have modified from previous reports. (**a**) In preliminary experiments, we confirmed that the device was rapidly cooled to −50 °C in vivo in mice at a 5 mm radius of the probe tip by thermal conduction through copper wires by pouring liquid nitrogen at −197 °C into the device. (**b**) The syringe was set on the manipulator (NARISHIGE, Tokyo, Japan). The manipulator enabled accurate probe insertion to the millimeter level.

**Figure 8 ijms-23-09445-f008:**
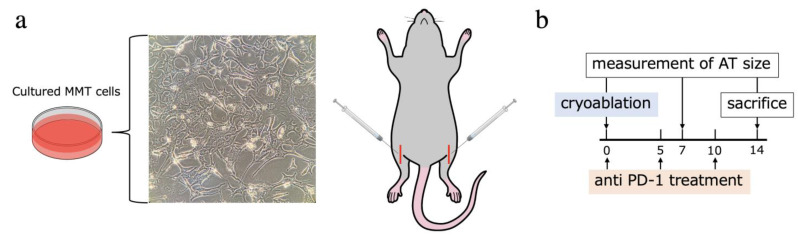
(**a**) Cultured MMT cells and schema of implantation of MMT suspended in Matrigel. (**b**) Schedule of treatment interventions and measurement of AT size. MMT, mouse mammary tumor; AT, abscopal tumor.

## Data Availability

All relevant data are within the paper.

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
