# Peer review of "Tumor-Specific Immunoenhancing Effects after Local Cryoablation for Metastatic Bone Tumor in a Mouse Model"

_ijms, 2022, doi:10.3390/ijms23169445_

Round 1

Reviewer 1 Report

Annen et al, manuscript entitled “Tumor-Specific Immunoenhancing Effects After Local 2 Cryoablation for Metastatic Bone Tumor in a Mouse Model” has demonstrated the abscopal effect of cryoablation associated with bone metastasis in combination with anti-PD1, also studied the immune response upon therapy,

 There is major concern of the study,

1)      The tumor growth inhibition was well reported in Fig.1

2)      It would be better, if display the spleen images Fig.2

3)      There are other key cytokines involved in regulation of PD-1/PD-L1, like IL-10, CXCR8, IL-6

4)      Its not precise to do immunostaining for T-cell function to check of CD4, CD8, FOXP3, it would be better if done FACS analysis

Thank you,

Reviewer 2 Report

Very interesting novel research potentially impacting clinical practice.

I have some suggestions:

- Please insert in the abstract briefly the main results (with statistical data).

- The paragraphs' order in the paper should be revised. Please move Methods before results.

- Talking about the advantages of cryoablation in clinical practice, please mention the 'ice-ball' : the real time assessment of the ablated areas.

- In both methods and results please insert and detail the number of subjects involved (mice) totally and in the different subgroups.

- Please underline and discuss briefly the limits of the study in a section into 'Discussion'. The only limitation reported by far is that your mouse model of bone metastasis was not a  bone metastasis model by strict definition.

- Please discuss briefly, in 'discussion' section if there are others researches on the abscopal effects of other thermal ablation techniques in oncology (cite and discuss)

- Please, discuss in depth future possible applications in clinical human research; which could be the 'first step'?

Round 2

Reviewer 1 Report

The manuscript was well improved, I recommend for consideration for publish in the International Journal of Molecular Sciences 

Reviewer 2 Report

I am satisfied with the revisions performed

congratulations for the research performed!